# Applying Multiple Functional Connectivity Features in GCN for EEG-Based Human Identification

**DOI:** 10.3390/brainsci12081072

**Published:** 2022-08-12

**Authors:** Wenli Tian, Ming Li, Xiangyu Ju, Yadong Liu

**Affiliations:** College of Intelligence Science and Technology, National University of Defense Technology, Changsha 410073, China

**Keywords:** EEG, GCN, human identification, functional connectivity, feature fusion

## Abstract

EEG-based human identification has gained a wide range of attention due to the further increase in demand for security. How to improve the accuracy of the human identification system is an issue worthy of attention. Using more features in the human identification system is a potential solution. However, too many features may cause overfitting, resulting in the decline of system accuracy. In this work, the graph convolutional neural network (GCN) was adopted for classification. Multiple features were combined and utilized as the structure matrix of the GCN. Because of the constant signal matrix, the training parameters would not increase as the structure matrix grows. We evaluated the classification accuracy on a classic public dataset. The results showed that utilizing multiple features of functional connectivity (FC) can improve the accuracy of the identity authentication system, the best results of which are at 98.56%. In addition, our methods showed less sensitivity to channel reduction. The method proposed in this paper combines different FCs and reaches high classification accuracy for unpreprocessed data, which inspires reducing the system cost in the actual human identification system.

## 1. Introduction

Due to the increasing security requirements for identification systems, biometric technologies have been extensively researched. Fingerprint, iris, face, and other biometric authentication methods have been extensively used in real life. However, these biometrics each possess their weaknesses [1]. For example, fingerprints can be easily stolen by touching the surface of an object. More importantly, these biometrics are non-renewable. Once the biometrics are stolen, the user cannot cancel them, as the user cannot simply change their face, iris, and fingerprints.

EEG is a recording of the electrical activity of the brain, and studies indicate that the EEG could be unique for each individual and thus has the potential to be a biometric [2]. EEG biometrics have unique advantages compared to traditional biometrics. Firstly, the acquisition of EEG cannot be performed without the users’ knowledge, therefore EEG biometrics are difficult to be stolen. Secondly, EEG biometric are reproducible. New EEG biometrics can be generated by simply changing the pattern of brain activity. Finally, EEG is sensitive to emotions such as stress, which makes it impossible to perform EEG identification in situations such as when the user receives coercion. For all these reasons, EEG-based identification holds the potential to be used in scenarios demanding a high security level [3].

Human identification is a prediction of a user’s identity among a group of people [4]. EEG-based human identification uses machine learning methods to classify EEG data from different people to identify different individuals. The most important thing in EEG-based identification is to select the EEG features that maximize the differences between different individuals [5]. EEG functional connectivity (FC) is a kind of index that describes the dependence and coupling relationship between different brain regions [6]. In the field of EEG analysis such as disease diagnosis, FC tools have been used to better observe the integration of distributed information of regular and synchronized multi-scale communication within and across inter-regional brain areas [7]. In previous studies, EEG-FC has been shown to have characterizing properties for different individuals [8]. Commonly used FC indicators in identity recognition are Pearson’s correlation coefficient (*COR*), coherence (*COH*) [9,10,11], phase lock value (*PLV*) [4,12,13,14,15,16], mutual information index (*MI*) [9], and Granger causality index (GC) [8], etc. These features are often used in isolation, which limits the accuracy improvement. When using delicate pre-processed data, the accuracy can reach 90.30% [8]–99.85% [4]. These FC indicators carry different information, which is complementary for the identification. Employing more FC features in identification may help to improve the accuracy. However, due to the redundancy and the unclear physiological mechanism of different FC features, efficiently integrating and utilizing these multiple features is challenging.

For traditional machine learning methods, simply incorporating more features will inevitably increase the dimension of the machine learning model. Especially when there is a lot of redundancy among different features, the model order is much higher than the real dimension of the feature, and this may result in overfitting. 

Graph convolutional neural network (GCN) is a method for processing graph signals. Different from general machine learning algorithms, GCN not only needs to specify the features to be processed (node signals) but also needs to specify the connection relation-ship between features (graph structure) according to graph structure to aggregate neighbor node signals. In GCN, the dimension of the training parameters only depends on the column of the node signals rather than the graph structure.

This inspires solving the problem of multi-feature utilization: using concatenated multiple FCs as the structure matrix of GCN and using a homogenized vector as the node signal of GCN to aggregate information of the FC matrix. Because a larger structure matrix does not lead to more training parameters, the above GCN network can incorporate multiple FC features without increasing the model order. 

In this work, a multiple-features integrating method based on GCN is proposed for identity recognition using multiple FC indicators. The new method was evaluated on a classic public dataset using different quantity of FCs and their different combination. The results showed that employing multiple FC features in GCN can improve the accuracy of the identification system compared to using a single FC feature. The best two-FC combination is *MI + PLV* with 98.51% accuracy, which is more than 3% better than the accuracy with a single FC. In addition, we evaluated the sensitivity of our method to channel reduction. The results showed that our method retained an accuracy of about 94.26% even when the number of channels was reduced to half of the original.

The rest of this paper is organized as follows: Section 2 presents the materials and a detailed approach to how multiple FC features can be used as graph structures. In Section 3, the evaluation results are reported. Section 4 explains the evaluation results and discusses the innovation and applicability of this work. Section 5 summarizes the work of this manuscript.

## 2. Materials and Methods

In this section, the dataset, FC computation, graph representation, and how our model is designed are introduced separately for this work.

### 2.1. Datasets and Preprocessing

We chose the widely used public dataset EEG Motor Movement/Imagery Dataset [17,18] to evaluate our method. This dataset comes from PhysioNet and contains more than 1500 one-minute and two-minute EEG recordings collected from more than 1000 subjects. Each subject performed 14 experiments: two one-minute baseline experiments with eyes open (EO) and eyes closed (EC), and three two-minute experiments with four motor or imagery tasks, four tasks, which are left or right fist open and closed (MOV-lr), imagine left or right fist open and closed (IMG-lr), double fist or both feet open and closed (MOV-b), imagine open and closed fists or feet (IMG-b). In this work, we used data from the baseline EO task for testing. The EEG was recorded from 64 electrodes as per the international 10–10 system (excluding electrodes A1 and A2). The sampling rate was 160 Hz. The description of Motor Movement/Imagery Dataset is illustrated in Table 1.

The main purpose of this work is to explore the classification capabilities of GCN networks for identity recognition when using multiple FCs as graph structures. To reduce the influence of irrelevant factors, only the sliding window method was adopted to segment the original EEG data, without any preprocessing methods such as removing bad segments that require manual participation. The sliding window used has a window length of 1 s and an overlap of 50%. Another reason to use sliding windows is for data augmentation.

### 2.2. Functional Connectivity Matrix

In this manuscript, four commonly used FCs are employed, namely Pearson’s correlation coefficient (*COR*), coherence (*COH*), phase locking value (*PLV*), and mutual information (*MI*). Detailed description and calculation process are listed below:Pearson’s correlation coefficient (*COR*)

*COR* is the simplest FC metric. It measures the time–domain linear correlation between two signals $x\left(t\right)$ and $y\left(t\right)$ at zero lag. For zero mean, the unit variance signal is defined as:(1)$$R_{xy}=\frac{1}{N}\overset{N}{\underset{k=1}{\sum }}x\left(k\right)y\left(k\right)$$

The value $R_{xy}$ range is [−1, 1], where a value of −1 represents a complete linear anti-correlation between the two signals, a value of 0 represents no linear dependence between the two signals, and a value of 1 represents a complete linear direct correlation between the two signals.

2.Coherence (*COH*)

*COH* measures the linear dependence of two variables $x\left(t\right)$ and $y\left(t\right)$ in the frequency domain. *COH* can be calculated as follows:(2)$$COH_{xy}\left(f\right)={\left|K_{xy}\left(f\right)\right|}^{2}=\frac{{\left|S_{xy}\left(f\right)\right|}^{2}}{S_{xx}\left(f\right)S_{yy}\left(f\right)}$$
where $S_{xy}\left(f\right)$ is the cross power spectral density between $x\left(t\right)$ and $y\left(t\right)$, $S_{xx}\left(f\right)$ and $S_{yy}\left(f\right)$ are the individual power spectral densities of $x\left(t\right)$ and $y\left(t\right)$. The value $COH_{xy}$ range is [0, 1]. A value of 0 means that the two signals have no linear correlation at the frequency f. A larger value means a stronger correlation between the two signals at the frequency f. A value of 1 means the correspondence between $x\left(t\right)$ and $y\left(t\right)$ at frequency f. Similar to *COR*, *COH* can only characterize the linear correlation between two signals.

3.Phase Locking Value (*PLV*)

*PLV* estimates how the relative phase is distributed over the unit circle. It is defined as:(3)$$PLV=\left|n^{-1}\overset{n}{\underset{t=1}{\sum }}e^{i\left(\phi_{xt}-\phi_{yt}\right)}\right|$$
where n represents the time point, $\phi_{xt}$ and $\phi_{yt}$ respectively represent the phase angles of the signals x and y at the time point t. The value range of *PLV* is [0, 1], and the larger the value, the stronger the phase synchronization between the two signals. The center frequency and bandwidth to be analyzed are usually specified in the *PLV* calculation and are usually divided according to the four bands of the EEG, that is, according to δ (1–3 Hz), θ (4–7 Hz), α (8–13 Hz), β (14–30 Hz) to calculate the *PLV* index.

4.Mutual Information (*MI*)

*MI* is an important indicator of information theory. EEG is essentially a multi-dimensional time series signal; thus, it is meaningful to measure EEG signals with the complexity indexes of information theory. Complexity measures of time series, such as entropy and Lempel–Ziv complexity, have been employed to quantify the dynamical changes of EEG [19,20,21,22,23,24]. Based on concepts from information theory, *MI* measures the interdependence between two variables, and it quantifies the amount of information that can be obtained about a random variable by observing another. In the EEG functional connection, the mutual information index can represent the amount of information contained in each other between the time-series signals of the two channels. *MI* can be calculated as:(4)$$MI_{xy}=\underset{i}{\sum }p\left(x,y\right)\log\frac{p\left(x,y\right)}{p\left(x\right)p\left(y\right)}$$
where $p\left(x,y\right)$ is the joint probability distribution function of $x\left(t\right)$ and $y\left(t\right)$. $p\left(x\right)$ and $p\left(y\right)$ are the marginal probability distribution functions of $x\left(t\right)$ and $y\left(t\right)$. This equation represents the cross mutual information between $x\left(t\right)$ and $y\left(t\right)$. The value range of *MI* is [0, 1]. When the value of *MI* is close to 1, it indicates that the two signals and the mutual information are completely correlated. When the value of *MI* is close to 0, it indicates that the two signals are completely independent.

### 2.3. Graph Representation

At present, the methods of analyzing EEG signals with GCN have been widely studied, and one of the basic tasks is to obtain the graph representation of multi-channel EEG signals. In this section, the general concepts of the graph and the graph representation of multiple FCs will be introduced.

#### 2.3.1. General Concepts of Graph

The graph consists of vertices and edges connecting the vertices, denoted as $G=\left(V,\;E\right)$, where V is the vertex set and E is the edge set. A graph with *n* nodes whose nodes $v_{i}\in V$, edges $e_{ij}\in E$, where $i,j\in \left\{1,2,\ldots ,n\right\}$.

A graph signal is a signal defined on a node of the graph, denoted as X, $X={\left[x_{1},x_{2},\ldots ,x_{n}\right]}^{T}$, where $x_{i}$ represents the signal strength on the node $v_{i}$. When analyzing the properties of the graph, in addition to the strength of the graph signal, it is also necessary to consider the topological structure of the graph. Based on the transformation of adjacency matrix A, the renormalized Laplacian matrix ${\tilde{L}}_{sym}$ is usually used to study the structural properties of the graph. The adjacency matrix $A={\left[a_{ij}\right]}_{m\times n}$ represents the connection relationship between nodes. ${\tilde{L}}_{sym}$ is defined as:(5)$${\tilde{L}}_{sym}={\tilde{D}}^{-1/2}\tilde{A}{\tilde{D}}^{-1/2}\;$$
where $\tilde{A}=A+I$, ${\tilde{D}}_{ii}=\sum_{j}{\tilde{A}}_{ij}$.

#### 2.3.2. Graph Representation of Multiple FCs

In this work, the EEG electrodes are regarded as nodes. The graphs defined by a single FC are called subgraphs. Multiple subgraphs are fused into graphs by different fusion strategies, add and concat. For add-fusion, the adjacency matrices of subgraphs are averaged. For concat-fusion, the adjacency matrices of subgraphs are concatenated.

Add-fusion

The add-fusion is used to average the signal matrices of subgraphs containing different types of FC matrices, and the adjacency matrices are also averaged:(6)$$\left\{\begin{array}{c} X_{add}=mean\left(X,X^{\prime },\ldots \right)\\ A_{add}=mean\left(A,A^{\prime },\ldots \right) \end{array}\right.$$
where $X_{add}$ and $A_{add}$ represent the signal strength matrix and adjacency matrix of the new graph obtained by add-fusion. X and A are the signal strength matrix and adjacency matrix of the first subgraph. X′ and A′ are the signal strength matrix and adjacency matrix of the second subgraph, and so on.

After the add-fusion, the signal strength of the graph and the connection strength of the graph structure are averaged, while the size of the adjacency matrix does not change; that is, no new connections are added.

2.Concat-fusion

The concat-fusion preserves the signal values of all original nodes, concatenates the adjacency matrices of multiple subgraphs along the diagonal, and then fills the corresponding positions with the relationship matrices of multiple subgraphs:(7)$$\left\{\begin{array}{c} X_{concat}={\left[X,X^{\prime },\ldots \right]}^{T}\\ A_{concat}=\left[\begin{array}{ccc} A & R & \ldots \;\\ R & A\prime & \\ \;\vdots & & \ddots \end{array}\right] \end{array}\right.$$
where $X_{concat}$ and $A_{concat}$ represent the signal strength matrix and adjacency matrix of the new graph obtained by concat-fusion. $R_{ij}\in R$ represents the relationship matrix between the subgraphs.

Taking the fusion of two subgraphs as an example, as shown in Figure 1, two subgraphs and their adjacency matrix of different FC are represented as blue (subgraph 1 and A) and yellow (subgraph 2 and A′). The nodes are $V=\left\{v_{a},v_{b},v_{c}\right\}$ and $V\prime =\left\{v_{a\prime },v_{b\prime },v_{c\prime }\right\}$, and the edges are $E=\left\{e_{ab},e_{ac},e_{bc}\right\}$ and $E\prime =\left\{e_{a\prime b\prime },e_{a\prime c\prime },e_{b\prime c\prime }\right\}$. A and A′ are the adjacency matrix within subgragh 1 and subgraph 2, respectively. The edges between the nodes from different subgraphs are colored in red. R is the relational matrix between the two subgraphs. In this work, we consider a simple case where the relational graph of two subgraphs is undirected. $A_{concat}$ is the adjacency matrix of the new graph obtained by concat-fusion. It can be seen from $A_{concat}$ that when the two subgraphs are combined by the concat-fusion, the number of nodes is doubled. The rows and columns of the adjacency matrix are doubled (the number of rows and columns of $A_{concat}$ in the graph changes to 6), and new connections appear between the nodes of the original two subgraphs (Matrix of the red part of the top right and bottom left corners of $A_{concat}$).

This work designs three types of relational matrices R for the concat method, which are zero matrix, identity matrix, and relational matrix based on Pearson correlation coefficient (concat_zero, concat_eye, and concat_corr).

The calculation method of the relationship matrix based on the Pearson correlation coefficient is:(8)$$r_{ij}=\left\{\begin{array}{c} \rho_{A\left(i,:\right)A\prime \left(j,:\right)}\;\;\;\;\;\;\;\;\;\;\;\;if\;i=j\\ 0\;\;\;\;\;\;\;\;\;\;\;\;\;\;\;\;\;\;\;\;\;\;\;otherwise \end{array}\right.$$
where $\rho_{xy}$ represents the Pearson correlation coefficient of the two variables x and y, $A\left(i,:\right)$ and $A\prime \left(j,:\right)$ represent the connection strength signals of nodes $v_{i}$ and $v_{j}$ to other nodes in the two subgraphs.

### 2.4. The Proposed Multi-FC Fusion GCN Model

The proposed multi-FC fusion GCN model is shown in Figure 2. Firstly, the original data are preprocessed and segmented. Then, different types of FC matrices are calculated as adjacency matrices. After that, we use the add or concat-fusion to combine multiple FCs and set the padding matrix to the three relationship matrices R described in Section 2.3.2. Through the above steps, we obtain a new adjacency matrix. Finally, a two-layer GCN network is employed to realize the classification of identities. The two layers of GCN are respectively defined as:(9)X1=σleakyReLUL˜sym,X0|W1,b1       W1∈ℝn×32,b1∈ℝ32X2=σtanhL˜sym,X1|W2,b2                W2∈ℝ32×8,b2∈ℝ8
where σ is the activation function. Wi and bi are weights and bias matrices of the ith layer. L˜sym is the renormalized laplacian matrix defined in Section 2.3.1. X is the signal strength matrix. In this work, the leakyReLU is applied to the first GCN layer with 32 convolutional neurons, and the tanh is applied to the second GCN layer with 8 convolutional neurons. For no other features except FC are employed, X0 is set to one matrix 1N×1.

After the two GCN layers, a dense layer is connected to map the feature space calculated by the previous layer to the sample label space, and finally, the softmax layer is used to realize the classification.

The model is written and implemented based on the python 3.7 and tensorflow 2.3.1 framework. NVIDIA GeForce RTX 2080Ti is used for model training. During training, a five-fold cross-validation method is used.

## 3. Results

### 3.1. The Evaluation Results by Using Different Combinations of Multiple FCs

In this work, we first evaluated the classification effect of four kinds of FC (*COR*, *COH*, *PLV*, and *MI*) used as adjacency matrices alone. The results are listed in Table 2. Then, we evaluated the classification accuracy of different FC combinations under different fusion methods (add, concat_zero, concat_eye, and concar_corr). There are a total of 11 combinations, and the detailed data are given in Table 3.

It can be seen from Table 2 that when the four FCs are used alone as the adjacency matrix of the graph, both *COR* and *PLV* achieve an accuracy of more than 95.26%, and *COH* alone achieves an accuracy of more than 91.21%. The worst performing *MI* can also achieve an accuracy of over 73.81%, which shows that the selected four FCs have a certain recognition ability for the identification task.

For convenience to compare the classification effect of single FC and multiple FCs, Figure 3 is reported, in which the left side shows the accuracy by using a single FC, and the right side shows the accuracy by using 11 different FC combinations with four different fusion methods.

As shown in Figure 3, in the case of two-FC combination, all four fusion methods are more effective than a single FC. The best two-FC combination is *MI+PLV* with the add-fusion, which achieves an accuracy of 98.51%. However, in the case of three-FC combination and four-FC combination, accuracy is not always higher than that in the two-FC combination. The accuracy of the combination *COR+COH+MI* with the add-fusion is 90.16%, which is lower than the combination *COH+COR* (96.36%), *MI+COR* (96.66%), and *MI+COH* (92.43%).

For further investigation of the phenomenon that the number of FCs increases while the accuracy decreases, Figure 4 is reported, where (a), (b), (c), and (d) show how the accuracy changes with the number of FCs in the four methods.

It can be seen in Figure 4a that under the add-fusion, when the number of FCs increases, the accuracy does not improve; the highest accuracy was achieved by two-FC combination. As shown in Figure 4b–d, in all three concat-fusions, the accuracy increases with the number of FCs. This may be due to the difference in the calculation method of add and concat, which is discussed in detail in the discussion section.

### 3.2. The Sensitivities to EEG Channel Numbers

In a human identification system, the number of channels is an important consideration. If the number of channels is too large, the deployment cost and complexity of the system will increase, which will limit the practical application [25]. Therefore, in the further study, we evaluate the performance of the proposed model in the case of channel reduction.

To evaluate the sensitivity of our method to the number of channels, we designed experiments to compare the classification accuracy by using single FC and multiple FCs at the different numbers of channels.

Firstly, we uniformly picked 32 channels from the original 64 channels by spatial distribution. Then, we evaluated the classification effect of single FC and multiple FCs of 32-channel-EEG. The results are shown in Figure 5, and detailed data are given in Table 4.

The left figure in Figure 5 shows the accuracy of four single FCs under 64 channels and 32 channels. It can be seen that when the number of channels is reduced to half of the original, the accuracy of the four single FCs decreases. *PLV* obtains the highest accuracy under 32 channels, reaching 82.58%.

The right figure in Figure 5 shows the accuracy of six two-FC combination under four different fusion methods under 32 channels. It can be seen that when the number of channels is reduced to half of the original, the accuracy of the FC combination using the concat-fusion is improved. Especially in the combination including *PLV* (*PLV+COR*, *PLV+COH,* and *MI+PLV*), the result is up to 94.26%, which is almost the same as the optimal single-channel under 64 channels. However, the result of using the add-fusion is not as good as a single FC, except for the combination *PLV+COR*.

In further exploration, to study the trend of performance when the number of channels is reduced, we give the accuracy of our method (*PLV+COR*) at a different level of channel reduction (64, 56, 48, 40, 32, 24, 20 and 16 channels remain); see Figure 6 and Table 5.

It can be seen from Figure 6 that the accuracy of both single FC and multiple FCs decreases as the number of channels decreases. The accuracy of multiple FCs with the concat-fusion declines slowly. When the number of channels is reduced to 40, the concat-fusion still maintains more than 97.13% accuracy, which is competent for the practical application of human identification. 

Compared with the three methods of concat, when the number of channels is greater than or equal to 32, the accuracy of concat_eye is slightly higher than the other two methods. When the number of channels is less than 32, all three methods show a decline, but the accuracy of concat_corr decreases more slowly, which remains at 63.19% under 16 channels. This indicates that concat_corr is better than concat_zero and concat_eye in the ability to resist channel reduction.

## 4. Discussion

Many works are adopting GCN with FC in graph structure to deal with the individual differences of brain networks. Behrouzi et al. [4] set the signal feature as vector 1 and used single *PLV* or *COR* as the graph structure of GCN. This proves the feasibility of using FC as a graph structure to achieve stable identity recognition. In this paper, we use fused multiple FCs as the structure matrix of graphs. Because a larger structure matrix does not lead to more training parameters, our method can incorporate multiple FC features without increasing the model order. 

The accuracy of using two-FC combination under 64 channels is up to 98.51% (*MI+PLV*). We used data without preprocessing. Most of the existing studies rely on delicate or even offline manual pre-processed data. With preprocessed data, the accuracy of the existing methods reached 90.30% (*MI*, *SVM* [8]), 98.75% (*COR* CNN [13]), and 99.85% (*PLV*, GCN [4]). Our method achieves an accuracy rate comparable to or exceeding existing methods even when using unpreprocessed data. This may be because our method reduces the effect of EEG artifacts by combining multiple FCs. The results show that our method has a higher classification capability for unpreprocessed data, which contributes to the cost reduction of the actual system deployment.

Based on the results under 64 channels reported in this paper, it can be seen that when a single FC could identify different people, the concat-fusion has a positive effect on improving the classification accuracy. With the increase in the number of multi-FC, the classification accuracy increases. This suggests that the concat-fusion can comprehensively utilize information of various types of FCs. Different from the concat-fusion, the evaluation of the add-fusion under 64 channels shows that in the case of two-FC combination, the add-fusion can bring some improvement to the accuracy. However, when the number of multiple FCs increases, the accuracy sometimes decreases, which is because the process of adding and averaging different types of FCs may cause a loss of features of conflicting changes between different FCs.

Based on the accuracy results of channel reduction reported in this paper, it can be seen that in the case of channel compression, as the number of channels decreases, the accuracy of the single FC, add and concat-fusions all decreases. Compared with a single FC and add, the three concat-fusions all decrease more slowly. This is because the graph structure obtained by the concat-fusion has more connections. When the number of channels is reduced (the number of nodes is reduced), more edge connections are beneficial to keep the aggregation path of information in the GCN network strong. In the comparison between the three concat-fusions, the concat_corr method shows higher accuracy when the number of channels is smaller (<32), which is because concat_corr artificially adds the relationship between the subgraphs of different FCs to express the relationship between each other. Compared to using zero matrix or eye matrix as the padding matrix, the subgraph correlation matrix itself also carries individual-specific information, which is conducive to further improve the classification ability.

Based on the above results, some application suggestions can be put forward for the actual deployment of the identification system. That is, using the concat-fusion to combine different types of FCs in the actual system can reduce the requirement for the minimum number of channels while ensuring a high accuracy rate. For example, when the dataset employed in this manuscript is used for identification, the accuracy of *PLV+COR* with the concat-fusion under 40 channels exceeds 97.13%, which is better than using any single FC alone under 64 channels This is of extraordinary significance for reducing the actual system cost.

## 5. Conclusions

At present, EEG identification using FC has become an extensively studied technique. However, how to apply multiple FCs in recognition is a challenging problem. This paper utilizes multiple FC features as the structure of the graph and explores the identification performance on a classical public dataset. The results show that utilizing multiple FCs can improve the accuracy of the identity authentication system, the best results of which are at 98.56%. In addition, this manuscript evaluates the sensitivity of our method to the number of channels. When the number of channels is reduced from 64 to 40, the combination *PLV+COR* with the concat-fusion maintains an accuracy of over 97.13%. This result suggests that multiple FCs can be adopted with the concat-fusion on fewer channels to reduce the system cost while maintaining high recognition performance, which inspires the actual deployment of the human identification system.

## Figures and Tables

**Figure 1 brainsci-12-01072-f001:**
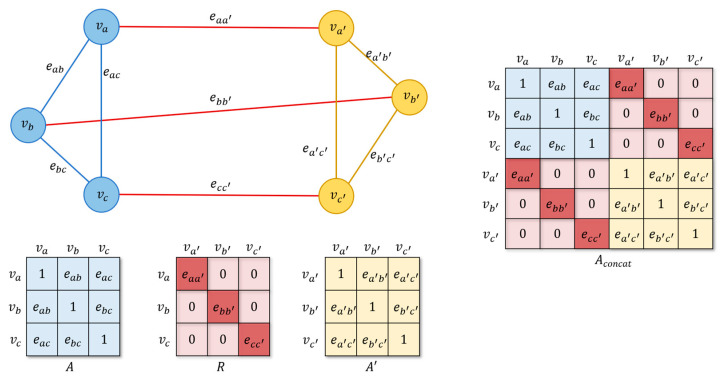
The schematic of the fusion of two subgraphs. Two subgraphs and their adjacency matrix of different FC are represented as blue (subgraph 1 and A) and yellow (subgraph 2 and $A^{\prime }$). A and $A^{\prime }$ are the adjacency matrices within subgragh 1 and subgraph 2, respectively. The edges between the nodes from different subgraphs are colored in red. R is the relational matrix between the two subgraphs. $A_{concat}$ is the adjacency matrix of the new graph obtained by concat-fusion.

**Figure 2 brainsci-12-01072-f002:**
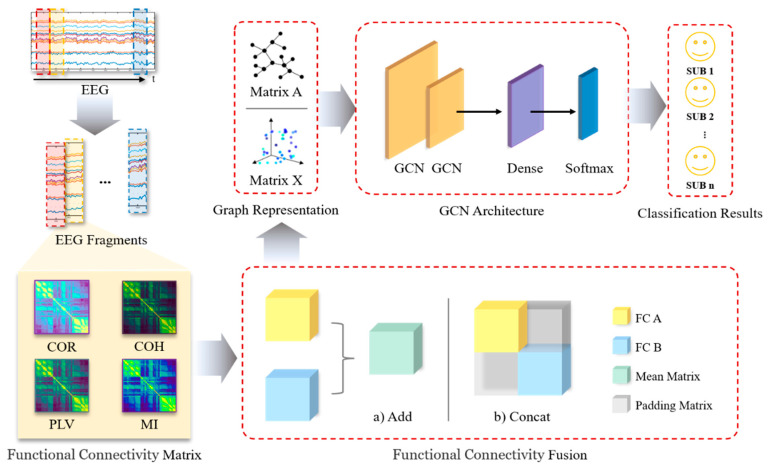
The flow chart of the proposed multiple features integrating method based on GCN.

**Figure 3 brainsci-12-01072-f003:**
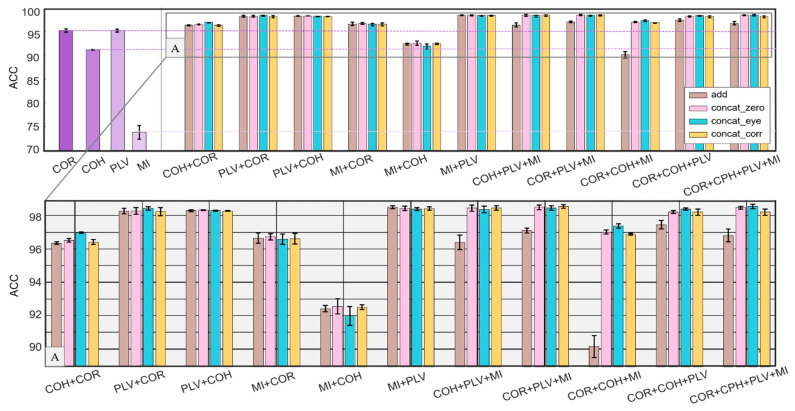
Comparison between using single FC and multiple FCs on classification accuracy. Upper left: accuracy by using single FC. Upper right: accuracy by using 11 different FC combinations with four different fusion methods. Below: the enlarged view of the rectangle in the upper right figure. It can be seen that all four fusion methods are more effective than a single FC in the case of two-FC combination. The best two-FC combination is *MI+PLV*, which achieves an accuracy of 98.51% with the add-fusion. However, in the case of three-FC combination and four-FC combination, accuracy is not always higher than that in two-FC combination. The accuracy of the combination *COR+COH+MI* with the add-fusion is 90.16%, which is lower than the combination *COH+COR* (96.36%), *MI+COR* (96.66%), and *MI+COH* (92.43%).

**Figure 4 brainsci-12-01072-f004:**
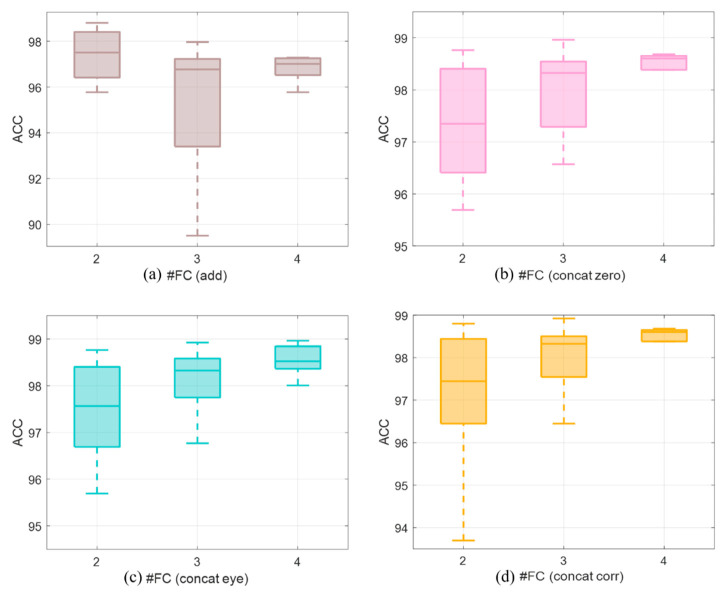
Comparison between using different quantities of FCs with the four fusion methods ((**a**): add, (**b**): concat_zero, (**c**): concat_eye, and (**d**): concat_corr). In the add-fusion, when the number of FCs increases, the accuracy does not improve; the highest accuracy was achieved by two-FC combination (subfigure **a**). In all three concat-fusions, the accuracy increases with the number of FCs (subfigures **b**–**d**).

**Figure 5 brainsci-12-01072-f005:**
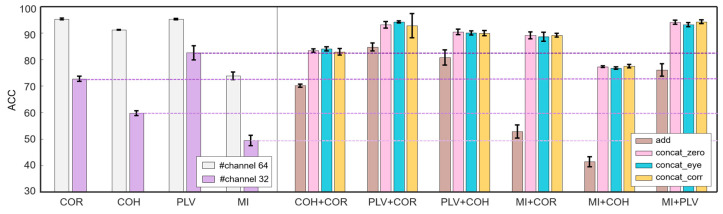
Classification accuracy comparison by using single FC and multiple FCs in the case of channel reducing. Left: accuracy by using four single FCs under 64 channels (grey bars) and 32 channels (purple bars). Right: accuracy by using six two-FC combination under 32 channels. When the number of channels is reduced to half of the original, the accuracy of the FC combination using the concat-fusion is higher than by using single FC. Especially in the combination containing *PLV* (*PLV+COR*, *PLV+COH*, and *MI+PLV*), the best result is up to 94.26%, which is almost the same as the optimal single channel under 64 channels.

**Figure 6 brainsci-12-01072-f006:**
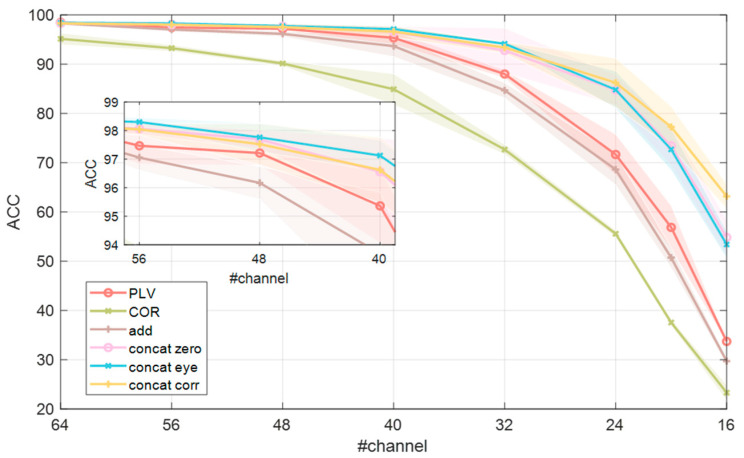
Performance comparison of different methods at the different levels of channel reduction (64, 56, 48, 40, 32, 24, 20, and 16 channels remain). It can be seen that the accuracy of multiple FCs with the concat-fusion declines slowly. When the number of channels is reduced to 40, the concat-fusion still maintains more than 97.13% accuracy. When the number of channels is greater than 32, the concat method shows a significant advantage.

**Table 1 brainsci-12-01072-t001:** Description of the motor movement/imagery dataset.

Motor Movement/Imagery Dataset			
#Channel *	64	Electrodes	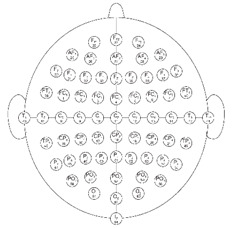
#Subject	109		
Fs (Hz)	160		
Tasks (duration × run)	EO (60 s)		
	EC (60 s)		
	MOV-lr (120 s × 3)		
	IMG-lr (120 s × 3)		
	MOV-d (120 s × 3)		
	IMG-d (120 s × 3)		

* #Channel means the number of channels. In the subsequent manuscript, the meaning of # is the same as here.

**Table 2 brainsci-12-01072-t002:** Classification accuracy by using single FC (*COR*, *COH*, *PLV*, and *MI*).

FC	*COR*	*COH*	*PLV*	** *MI* **
ACC/%	95.27	91.21	95.26	73.81

**Table 3 brainsci-12-01072-t003:** Classification accuracy by combining multiple FCs with different fusion methods.

#FC	Combination	Fusion Method (%)			
		Add	Concat_Zero	Concat_Eye	Concat_Corr
2	*COH*+*COR*	96.36	96.53	**96.99**	96.42
	*PLV*+*COR*	98.28	98.29	**98.44**	98.25
	*PLV*+*COH*	98.31	**98.34**	98.31	98.28
	*MI*+*COR*	96.66	**96.75**	96.59	96.62
	*MI*+*COH*	92.43	**92.56**	92.00	92.52
	*MI*+*PLV*	**98.51**	98.45	98.40	98.43
3	*COH*+*PLV*+*MI*	96.40	**98.47**	98.37	**98.45**
	*COR*+*PLV*+*MI*	97.10	98.51	98.46	**98.56**
	*COR*+*COH*+*MI*	90.16	97.02	**97.38**	96.89
	*COR*+*COH*+*PLV*	**97.46**	98.22	**98.40**	98.22
4	*COR*+*CPH*+*PLV*+*MI*	96.82	98.48	**98.56**	98.48

**Table 4 brainsci-12-01072-t004:** Accuracy of two-FC combination in different fusion methods under 32 channels.

#FC	Combination	Fusion Method (%)			
		Add	Concat_Zero	Concat_Eye	Concat_Corr
2	*COH*+*COR*	70.14	83.41	**84.04**	82.90
	*PLV*+*COR*	84.71	92.80	**94.17**	93.39
	*PLV*+*COH*	80.79	**90.44**	90.08	89.96
	*MI*+*COR*	52.80	89.08	88.66	**89.19**
	*MI*+*COH*	41.44	77.31	76.77	**77.51**
	*MI*+*PLV*	76.03	94.06	93.13	**94.26**

**Table 5 brainsci-12-01072-t005:** Accuracy of single FC and multiple FCs under different number of channels.

#Channel	Fusion Method (%)					
	*PLV*_Single	*COR*_Single	Add	Concat_Zero	Concat_Eye	Concat_Corr
64	95.26	95.27	98.28	98.32	**98.44**	98.29
56	94.16	93.27	97.06	98.07	**98.30**	98.05
48	90.95	90.16	96.17	97.69	**97.77**	97.53
40	88.02	84.93	93.69	96.56	**97.13**	96.63
32	82.58	72.70	84.71	92.80	**94.17**	93.39
24	71.68	55.59	68.63	84.76	84.83	**86.25**
20	56.88	37.57	50.74	73.45	72.72	**77.27**
16	33.72	23.33	29.72	54.85	53.41	**63.19**

## Data Availability

Not applicable.

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
