# Peer review of "Applying Multiple Functional Connectivity Features in GCN for EEG-Based Human Identification"

_brainsci, 2022, doi:10.3390/brainsci12081072_

Round 1

Reviewer 1 Report

I'm not sure if the number of decimal places in results is really necessary, and why it varies from 3 in abstract (98.556%), zero in Introduction ("about 95%"), 4 in Results (for example 96.6574%) and 3 (and zero again) in Conclusions (about 95%). This is a bit inconsistent and slightly blurs the presentation. Moreover, numbers in the Table 3 are not aligned to decimal point, making it harder to read. In my opinion numbers rounded to one (max. two) decimal places would be enough, because the difference between 0,0001 and 0,0002% is negligible, or at least it should be explained.

The article would benefit from the extension of the implementation information. The reader gets information about Python and tensorflow without any details. Are the versions of the Python important and why? Is the Author's code available (open source)?

There is no need for Appendix A, the tables A1 and A2 should be placed in the appropriate places within the main text. This way the whole content would become easier to follow.

Author Response

Thank you very much for your time involved in reviewing the manuscript and your very encouraging comments on the merits. We appreciate your clear and detailed feedback and hope that the explanation has fully addressed all of your concerns. In the following section, we discuss each of your comments individually along with our corresponding responses.

Point 1: I'm not sure if the number of decimal places in results is really necessary, and why it varies from 3 in abstract (98.556%), zero in Introduction ("about 95%"), 4 in Results (for example 96.6574%) and 3 (and zero again) in Conclusions (about 95%). This is a bit inconsistent and slightly blurs the presentation. Moreover, numbers in the Table 3 are not aligned to decimal point, making it harder to read. In my opinion numbers rounded to one (max. two) decimal places would be enough, because the difference between 0,0001 and 0,0002% is negligible, or at least it should be explained.

Response 1: Thank you for the detailed review. There are more decimal places in the table because we want to make it more accurate. However, as you mentioned, the excessive number of data bits seems unnecessary and makes reading more difficult, so I rounded all the data in the text to two decimal places to increase the overall readability.

Point 2: The article would benefit from the extension of the implementation information. The reader gets information about Python and tensorflow without any details. Are the versions of the Python important and why? Is the Author's code available (open source)?

Response 2: Thanks for your valuable suggestion. The original intention of giving versions of Python and tensorflow was to make it easier to reproduce the code, as we have not yet tested under other versions of Python and tensorflow and are not sure if unpredictable errors will arise under different versions. The code is currently being organized, it will be open source, and will be supplemented with tests on more versions of Python and tensorflow in the future.

Point 3: There is no need for Appendix A, the tables A1 and A2 should be placed in the appropriate places within the main text. This way the whole content would become easier to follow.

Response 3: Thanks for your great suggestion on improving the readability of our manuscript. We have adjusted the position of the tables as you suggested to make the whole content easier to understand, see table 4 and table 5.

We would like to take this opportunity to thank you for all your time involved and this great opportunity for us to improve the manuscript. We hope you will find this revised version satisfactory.

Reviewer 2 Report

July 26, 2022

The paper is interesting but I have the following comments and concerns:  

A. In the Abstract there should be pointed out (stressed) the general conclusion which follows from the conducted analysis. Some importance/consequences of this conclusion could be included.

B. In subsection: 2.1 datasets and preprocessing, the Authors should add the information that datasets come from PhysioNet.

C. A paragraph concerning the application of the Information Theory based method could be added, like in the following papers:

R. Hornero, D. Abásolo, J. Escudero, C. Gómez, Nonlinear analysis of elec-troencephalogram and magnetoencephalogram recordings in patients with Alzheimer’s disease, Philos. Trans. R. Soc. A 367 (2009) 317–336

H. Adeli, S. Ghosh-Dastidar, Automated EEG-based Diagnosis of Neurological Disorders - Inventing the Future of Neurology, CRC Press, Taylor and Francis, Boca Raton, Florida, 2010

Z.L. Bai, X. Li, A permutation Lempel–Ziv complexity measure for EEG analysis, Biomed. Signal Process. Control. 19 (2015) 102–114 

M. Bachmann, L. Päeske, K. Kalev, K. Aarma, A. Lehtmets, P. Ööpik, J. Lass, H. Hinrikus, Methods for classifying depression in single channel EEG using linear and nonlinear signal analysis, Computer Methods and Programs in Biomedicine 155 (2018) 11–17 

A. Pregowska, K. Proniewska K., P. M. van Dam, J. Szczepanski, Using Lempel-Ziv complexity as effective classification tool of the sleep-related breathing disorders, Computer Methods and Programs in Biomedicine 182 (2019)105052-1-7

D. Bajić, V. Đajić, B. Milovanović, Entropy Analysis of COVID-19 Cardiovascular Signals. Entropy 23 (2021)87

D. The quantitative comparison with other existing literature algorithms (papers) should be added in Section: Discussion, especially in taking into account accuracy. Maybe in the form of a Table.

E. The language should be carefully revised.

Final Comments 

The idea presented and developed in the paper seems to be interesting and the results obtained are promising (i.e. they are of some value), but due to the above concerns, I could not recommend this paper for publication (in the present form) provided that the above problems/questions will be carefully addressed and clarified. At this moment I would recommend Major Revision.

Author Response

We appreciate your clear and detailed feedback. We have responded to each of your comments. Please see the attachment for details. We hope that the explanation has fully addressed all of your concerns.

Reviewer 3 Report

The authors proposed a system for human identification based on a graph convolutional neural network which combines multiple functional connectivity features for classification. The results showed that the best accuracy was achieved by the combination of Phase Locking Value and Pearson’s correlation coefficient. Moreover, it was found that the accuracy maintained high values when the number of EEG channels was reduced to half the original number. The topic is interesting and the paper is well structured, but the Results section needs to be revised to improve readability. I have the following comments and suggestions:

In the introduction, add information about the results obtained in the cited papers [6-16].

Add the parameters used for the implementation of the GCN layers.

There are some discrepancies between the description of the results and the values given in the Tables:

- In Table 3, the accuracy value of 98.5082% is associated with MI+PLV, whereas in line 246 this value is associated with PLV+COR. Which one is correct?

- In lines 292-293 you wrote: “PLV obtains the highest accuracy under 32 channels, reaching 88.01%”, but in Table A2 this value is associated with 40 channels.

Author Response

(The authors gave the same response as above.)

Reviewer 4 Report

The article proposes an EEG-based person identification system based on functional connectivity measure and a graph convolutional neural network in the paper "Applying Multiple Functional Connectivity Features in GCN for EEG-based Human Identification" (GCN).

I have only a few concerns about the research that was done for this manuscript.

·      The notion of "EEG-based human identity," as the author states, is really not well addressed.

·      The statement "Among them, EEG-based identification technology has gained widespread attention due to its concealment, revocability, and problems in theft" must also be clarified, in my opinion.

·      I would advise refocusing the introduction and improving how the primary issue is handled.

· "Datasets and Preprocessing" section does not outline any preprocessing step. I would suggest going into great depth about the preprocessing methods employed. Using volume conduction mitigation strategies like Laplacian filters is something else I would suggest.

·      The author employs two well-known, yet distinct functional connections that are vulnerable to volume conduction. How the authors approached this problem is not obvious. I was curious as to how and in which bands the coherence was determined. It would be interesting to know what the authors performed to remove the bias caused by common EEG artifacts.

·      I'd like to understand what " and the signal ? on the nodes  is defined as the constant 1. Multiple FCs are used as the adjacency matrix ? to form a  graph, and the graph to be fused is called a subgraph” means?

·     The author of this paper computes the adjacency matrix in order to implement a graph, although this topic is not adequately covered. I'd like to suggest “A Machine Learning Approach Involving Functional Connectivity Features to Classify Rest-EEG Psychogenic Non-Epileptic Seizures from Healthy Controls” for preprocessing, feature dataset organization and implementation.

·      In “It can be seen that when the two subgraphs are combined by the concat-fusion, the number of nodes is doubled, while the signal value of the original node remains unchanged. The rows and columns of the adjacency matrix are doubled, and new  connections appear between the nodes of the original two subgraphs.” the meaning is not clear. We don't know how the authors construct the adjacency matrix for each connectivity measure. The graph appears to be constructed by concatenating the adjacency matrices after that. I would advise reporting literature for this strategy.

·      I would advise performing a grammar and language check.

·      I would advise fixing the y-scale across all figures so that findings may be compared.

·      It's probable that using this concatenation approach, one connection measure will influence the others; nonetheless, this study doesn't care.

·      It is unclear what kind of scientific methodology the authors followed to downsample the EEG layout (from 64 to 32).

·      I would advise a feature description and organization of the dataset as the authors have utilized several approaches to feed the convolution graph network.

Author Response

(The authors gave the same response as above.)

Round 2

Reviewer 2 Report

The authors considered all my comments. I recommend this paper for publication.

Author Response

Thank you for your time and effort in reviewing the manuscript, and your meticulous and enlightening suggestions on improving the paper.

Reviewer 3 Report

The authors addressed the issues I raised and the paper improved. 

Author Response

(The authors gave the same response as above.)

Reviewer 4 Report

The authors did a good revision, they boost the manuscript's scientific sound. 

I have only one additional suggestion.

Figure 5 is redundant, resulting in overlapp with that in table 1, I would suggestin to prune it.

Author Response

Thank you for your time and effort in reviewing the manuscript, and your meticulous and enlightening suggestions on improving the paper.

We have deleted Figure 5 and its related descriptions in the text. After that, the serial number of the caption in the text was updated.